# Heterogeneous Habitats in Taiga Forests with Different Important Values of Constructive Species Changes Bacterial Beta Diversity

**DOI:** 10.3390/microorganisms11102609

**Published:** 2023-10-23

**Authors:** Tian Zhou, Song Wu, Hong Pan, Xinming Lu, Jun Du, Libin Yang

**Affiliations:** 1Key Laboratory of Biodiversity, Institute of Natural Resources and Ecology, Heilongjiang Academy of Sciences, Harbin 150040, China; 15894479670@163.com (T.Z.); panhong500@163.com (H.P.); luxinming0210@163.com (X.L.); 2Science and Technology Innovation Center, Institute of Scientifc and Technical Information of Heilongjiang Province, Harbin 150028, China; wusong0927@126.com; 3Heilongjiang Huzhong National Nature Reserve, Huzhong 165038, China; jxnl0123@163.com

**Keywords:** Great Khingan Mountains, *Larix gmelinii*, soil bacteria, beta diversity

## Abstract

As a crucial link between the aboveground and belowground components of forest ecosystems, soil bacterial communities are extremely sensitive to changes in plant communities and soil conditions. To investigate the impact of the difference of constructive species on soil bacterial communities in taiga forests, we conducted a vegetation survey at the international monitoring plot of the *Larix gmelinii* forests in the Great Khingan Mountains and calculated the important value of *Larix gmelinii* to determine experimental groups based on this survey. Subsequently, we collected soil samples for high-throughput sequencing to analyze how the soil bacterial community composition and diversity changed, and which factors affected them. The results showed that taiga forests with different important values of *Larix gmelinii* had heterogeneous habitats, in which the soil AP content significantly increased, and the SOC, MBC, pH, and C/N content decreased significantly (*p* < 0.05). A total of 32 phyla, 91 classes, 200 orders, 308 families, 496 genera, and 975 species of soil bacteria were obtained by sequencing. Among them, Proteobacteria, Actinobacteriota, and Acidobacteriota were the dominant phyla, and *Mycobacterium* was the dominant genus, and the relative abundance of each bacterial group was varied. The beta diversity of soil bacteria showed extremely significant differences (*p* = 0.001), with SOC, C/N, MBC, AP, TN, pH, AN, and WC being the main influencing factors. Functional prediction analysis showed that chemoheterotrophy and aerobic chemoheterotrophy were the main bacterial functional groups, and the relative abundance of each functional group was significantly different (*p* < 0.05). Overall, taiga forests with differences in constructive species had heterogeneous habitats, which changed the community composition, beta diversity, and potential functions of soil bacteria.

## 1. Introduction

Soil microorganisms, as the most active biological components in forest ecosystems, contribute to many key ecological processes such as carbon, nitrogen, and phosphorus biochemical cycling, soil formation, and pollutant degradation [1,2,3]. Soil bacteria make up the majority of soil microorganisms, accounting for more than 90%, and their diversity and community composition are highly sensitive to environmental changes [4]. As well as being influenced by factors such as soil pH, moisture conditions, and nutrient content, soil bacterial communities will also show noticeable changes when aboveground plant communities shift [5,6]. A growing interest in ecology has been devoted to the relationship between soil bacterial communities and aboveground plant communities in recent years [7].

As part of forest ecosystems, there is a close interaction between soil bacteria and plants. The growth, development, and distribution patterns of plant communities are influenced by soil bacteria through the regulation of soil nutrient cycling [8]. On the other hand, it has been observed that plants have the ability to impact the composition and abundance of soil bacteria through several processes such as root activity, root exudates, and the features of litter [9,10,11]. Previous investigations have indicated that there are notable differences in the structure and function of soil bacterial communities across various vegetation types, as well as dramatical shifts in their composition over different stages of succession [12,13]. Furthermore, an extensive quantity of studies has provided evidence showing that the association between soil pH, soil carbon, nitrogen, phosphorus, and soil bacterial communities might exhibit positive, negative, or neutral correlations within diverse vegetation situations [14,15]. Therefore, changes in aboveground plant communities can lead to variations in soil bacterial communities, and the heterogeneity of the habitat where the vegetations are located has diverse impacts on the composition and diversity of soil bacterial communities. Much research has been conducted regarding the interactions between aboveground plant communities and soil bacterial communities. However, forest ecosystems, which constitute the primary component of terrestrial ecosystems, display notable differences in habitat within a specific region. These variations contribute to the complexity of the interactions between plant communities and soil bacteria. Consequently, further investigation is required to gain a deeper understanding of these interactions [16].

The region known as the Great Khingan Mountains lies in the northeastern part of China and represents the most widespread cold temperate coniferous forest environment inside the nation [17]. The particular forest community structure of the area can be attributed to its unique geological and climatic conditions, resulting in a composition dominated by a single plant species and exhibiting low diversity in the tree layer. There are a few tree species, including *Larix gmelinii*, *Betula platyphylla*, and *Populus davidiana*. Notably, *Larix gmelinii* holds the prominent position as the dominating constructive species within this area [18]. Although several studies have been carried out on the soil bacterial community in the Great Khingan Mountains during recent years, most of them have concentrated on climate change, forest types, and environmental disturbances [19,20,21]. There are few studies on the interaction between plant diversity and soil microbial diversity; particularly, whether there are differences in soil bacterial diversity after changes in the community structure of taiga forests due to the difference of constructive species has not been reported. Therefore, this study analyzed the vegetation within a 25 hm^2^ monitoring plot in cold temperate coniferous forests and estimated the important value of *Larix gmelinii* to reflect the differences of constructive species. The soil of taiga forests with different important values was taken as the study object, and high-throughput sequencing technology was employed to examine the difference of community composition and diversity of soil bacteria, as well as the factors influencing these variations. The findings of this study can serve as a reference and foundation for understanding the mechanisms underlying the construction and maintenance of bacterial diversity in soil.

## 2. Material and Method

### 2.1. Site Description

The research area is situated in the Huzhong National Nature Reserve, Daxinganling, Heilongjiang Province, China (51°49′01″ N–51°49′1″ N, 122°59′33″ E–123°00′03″ E; Figure 1). The terrain of the reserve is gentle, with an altitude of 847–974 m. It falls within the typical cold temperate continental monsoon climate zone, which is distinguished by substantial fluctuations in temperature throughout the year. The hottest month has an average temperature of 24.5 °C, while the coldest month has an average temperature of −35.8 °C. The average annual precipitation is 458.3 mm, and the majority of precipitation occurs from June to August, making up about 65% of the annual total. The composition of plant community is relatively straightforward, with *Larix gmelinii* being recognized as the primary constructive species in cold temperate coniferous forests [22].

### 2.2. Experimental Design and Soil Sample Collection

The experimental plot was a typical *Larix gmelinii* forest in Huzhong National Nature Reserve, and the quadrats of 20 m × 20 m were set there. The diameter at breast height, tree height, and number of trees were recorded within each quadrat. According to Formula (1), the important value of the constructive species (*Larix gmelinii*) was estimated [23]. Its calculation formula is as follows:(1)IV=(RD+RP+RH)/3
where IV means important value of *Larix gmelinii*; RD indicates relative density, the ratio of *Larix gmelinii* density to all species in the sample plot; RP represents relative dominance, the ratio of the basal area at breast height of *Larix gmelinii* to all species in the sample plot; RH means relative height, the ratio of the height of *Larix gmelinii* to all species in the sample plot.

According to the calculation results, the samples with significant differences in the important values of *Larix gmelinii* were selected as experimental groups, while those without significant differences were treated in the same experimental groups. A total of 5 distinct experimental groups were set up, comprising a total of 15 quadrats. The quadrats were sorted according to their important value as L1 < L2 < L3 < L4 < L5, as shown in Table 1.

The research applied the five-point sampling approach to collect soil samples from a depth of 0–20 cm after removing surface vegetation and covers. The soil samples were placed in sterile bags and brought back to the lab in ice boxes. The soil samples were divided into two portions, one of which was utilized to determine the soil physicochemical properties and the other for high-throughput sequencing of bacteria.

### 2.3. Characterization of Soil Physicochemical Properties

The total nitrogen (TN) and soil organic carbon (SOC) were quantified using an elemental analyzer (Multi N/C 2100S, Analytik Jena AG, Jena, Germany). Soil available potassium (AK) content was examined by the flame photometric approach. The quantity of soil available phosphorus (AP) was assessed using the molybdenum antimony colorimetric method beyond sodium bicarbonate extraction. Soil available nitrogen (AN) content was identified by the alkali hydrolysis diffusion method. Soil microbial carbon (MBC) content has been determined with the chloroform fumigation extraction method. The pH of the soil (soil-to-water ratio of 1:10) was measured by a pH meter (PB-10, Sartorius, Gottingen, Germany). The soil moisture (WC) content was obtained based on the aluminum box drying method [24,25].

### 2.4. DNA Extraction, PCR Amplification, and Illumina MiSeq Sequencing

Total microbial DNA was extracted from soil samples with the E.Z.N.A.^®^ Power Soil DNA Kit (Omega Bio-tek, Norcross, CA, USA). The concentration and purity of DNA was initially assayed by 1% agarose gel electrophoresis and then determined by a NanoDrop 2000 spectrophotometer (Thermo Fisher Scientific, Wilmington, DE, USA). The V3-V4 region of the bacterial 16S rRNA gene was amplified by using the universal primers 338F (5′-ACTCCTACGGGAGGCAGCA-3′) and 806R (5′-GGACTACHVGGGTWTCTAAT-3′). There were 4 μL TransStart FastPfu buffer, 2 μL dNTPs (2.5 mM), 0.8 μL each of forward and reverse primers (5 μM), 0.4 μL TransStart FastPfu DNA polymerase, 10 ng of DNA template, and enough ultrapure water in a 20 μL PCR amplification system. The PCR cycle was performed at 95 °C for 3 min (initial denaturation), followed by 27 cycles of 95 °C for 30 s (denaturation), 55 °C for 30 s (annealing), and 72 °C for 45 s (extension). The PCR products were detected by 2% agarose gel electrophoresis, further purified with AxyPrep DNA gel extraction kit (Axygen Biosciences, Union City, CA, USA) and then quantified using the QuantiFluor™-ST (Promega, Madison, WI, USA). The Illumina MiSeq high-throughput was entrusted to Majorbio Bio-Pharm Technology Co., Ltd. (Shanghai, China). The raw sequencing data were compiled by the NCBI SRA database under the accession number PRJNA828368.

### 2.5. Bioinformatics Analysis and Statistical Processing

Fastp software (0.19.6) was utilized to filter out the low-quality and ambiguous base sequences, and FLASH (v1.2.11) was used for splicing. DADA2 in Qiime2 software (2020.2) carried out the noise reduction and chimera removal, and amplicon sequence variants (ASVs) were acquired for analysis from each sample. According to the species annotation database (silva138/16s_bacteria), the naive Bayes classifier of Qiime2 was used to obtain the species taxonomic information of ASV by flattening with the minimum number of sample sequences.

Soil physicochemical properties among taiga forests were compared by one-way analysis of variance (ANOVAs) using SPSS-25.0. The Venn diagram at the ASV level was plotted using the ‘stat’ package of R-3.3.1, and histograms of soil bacterial communities at the phylum and genus level were plotted using Origin. Alpha diversity indices, including Coverage index, Chao1 index, Shannon index, and Simpson index, were analyzed by Mothur 1.0.2. The beta diversity of soil bacterial communities was compared using non-metric multidimensional scaling (NMDS) based on Bray–Curtis distance in R-3.3.1. To explore the association between soil bacterial community structure and soil physicochemical factors, RDA analysis was performed using the ‘vegan’ package of R-3.3.1. The correlation between the top ten bacterial species and soil physicochemical factors was analyzed by using R-3.3.1 software based on Pearson correlation coefficient. The potential ecological functions of soil bacteria were analyzed using the FAPROTAX package of R-3.3.1.

## 3. Results

### 3.1. Physicochemical Characteristics of Soil in Taiga Forests with Different Important Values

As shown in Table 2, soil AP content increased significantly with the increase in *Larix gmelinii* important value, and soil pH, SOC, MBC, and C/N decreased significantly (*p* < 0.05). Whereas TN, AN, WC, and AK contents did not show regular changes with the change of important values, they were also significantly different among groups (*p* < 0.05). Thus, it was evident that the soil physicochemical properties of taiga forests with different important values differed significantly.

### 3.2. Distribution Characteristics of Soil Bacterial Community

The total number of bacterial ASVs acquired from each soil sample was ranked as L5 > L3 > L2 > L4 > L1, with respective counts of 1734, 1617, 1581, 1529, and 1433 (Figure 2). The total number of ASVs shared by each group was 206, while the unique number of ASVs was 847 in L5, 738 in L1, 559 in L3, 548 in L2, and 513 in L4 (Figure 2).

The detected ASVs were divided into 32 phyla, 91 classes, 200 orders, 308 families, 496 genera, and 975 species based on their taxonomic classification. There were 12 bacterial phyla with relative abundances above 1% (Figure 3). Proteobacteria, Actinobacteriota, and Acidobacteriota were the dominant phylum [26], accounting for 76.43–83.31% of the total number of bacteria, but their relative abundance was not significantly different among groups (*p* > 0.05). Bacteroidota, Chloroflexi, Firmicutes, Myxococcota, Gemmatimonadota, WPS-2, and Desulfobacterota all showed significant differences among the groups (*p* < 0.05). The relative abundance of Chloroflexi increased significantly with the increase in important value of *Larix gmelinii*, while that of Bacteroidota and Myxococcota declined significantly.

There were 19 bacterial genera with relative abundance greater than 1% at the genus level, and *Mycobacterium* (3.79–5.90%) showed a higher relative abundance in each group, which was the dominant bacterial genus in taiga forests (Figure 4). Except for *Candidatus Udaeobacter*, the relative abundance of each genus was significantly varied (*p* < 0.05). Among them, *Bradyrhizobium* had a tendency to increase significantly with the *Larix gmelinii* important values, while *Bryobacter* had a tendency to decrease significantly.

### 3.3. Changes of Bacterial Alpha Diversity

The coverage index of each group was over 0.999, and no significant difference was found (*p* > 0.05, Table 3), indicating that the sequencing results were reasonable and could better reflect the actual situation of bacterial communities. Chao1 index was not significant difference between the groups (*p* > 0.05). Shannon index was the highest in L5 and significantly greater than L1 and L4 (*p* < 0.05). In contrast, the Simpson index was the lowest in L5 and significantly lower than L1 (*p* < 0.05), while there was no significant difference between the other groups (*p* > 0.05).

### 3.4. Changes of Bacterial Beta Diversity

The stress value was less than 0.05, indicating that NMDS analysis accurately reflected the degree of difference among samples (Figure 5). L3 and L4 were dispersed in the first quadrant, L2 in the second quadrant, L1 in the third quadrant, and L5 in the fourth quadrant, demonstrating that the bacterial community structure of L3 and L4 was similar but distinct from that of L1, L2, and L5. Based on Adonis analysis, it can be concluded that the beta diversity of soil bacteria in taiga forests with different important values shows extremely significant differences (*p* = 0.001).

### 3.5. Correlation Analysis between Soil Physicochemical Factors and Soil Bacterial Community

According to the results of RDA analysis (Figure 6), RDA1 and RDA2 described 54.03% and 10.61% of the discrepancies in soil bacterial community structure in taiga forests, respectively, with a cumulative interpretation rate of 64.64%. Among them, L1 and L2 were positively correlated with pH, C/N, SOC, and MBC, while negatively correlated with TN, AP, AN, AK, and WC. L3, L4, and L5 were positively correlated with TN, AP, AN, AK, and WC, while negatively correlated with pH, C/N, SOC, and MBC.

It can be seen from Table 4 that the degree of the influence of soil physicochemical factors on soil bacterial community structure was ranked as follows: SOC > C/N > MBC > AP > TN > pH > AN > WC > AK. Except for AK, SOC, C/N, MBC, AP, TN, pH, AN, and WC had significant effects on bacterial community structure (*p* < 0.05).

At the phylum level (Figure 7a), Actinobacteriota, Chloroflexi, and Gemmatimonadota were significantly positively correlated with TN (*p* < 0.01), AP (*p* < 0.001), and AN (*p* < 0.05) and significantly negatively correlated with SOC (*p* < 0.001), MBC (*p* < 0.01), pH (*p* < 0.05), and C/N (*p* < 0.001); Acidobacteriota was positively correlated with SOC (*p* < 0.05), MBC (*p* < 0.05), and C/N (*p* < 0.05) and negatively correlated with AP (*p* < 0.05); Bacteroidota and Myxococcota were significantly positively correlated with SOC (*p* < 0.001), MBC (*p* < 0.001), pH (*p* < 0.01), and C/N (*p* < 0.001) and significantly negatively correlated with TN (*p* < 0.01), AP (*p* < 0.001), AN (*p* < 0.01), and WC (*p* < 0.05); Firmicutes was significantly positively correlated with AP (*p* < 0.05) and significantly negatively correlated with MBC (*p* < 0.05); at the genus level (Figure 7b), *Bradyrhizobium* and *Candidatus Udaeobacer* were significantly positively correlated with TN, AP, and AN (*p* < 0.05) and significantly negatively correlated with SOC, MBC, pH, and C/N (*p* < 0.05); *Bryobacter*, *Granulicella*, *Burkholderia Caballeronia Paraburkholderia*, and *Mucilaginibacter* were significantly positively correlated with SOC, MBC, pH, and C/N (*p* < 0.05) and highly significantly negatively correlated with TN, AP, and AN (*p* < 0.05). *Roseiarcus* was significantly positively correlated with SOC (*p* < 0.001), MBC (*p* < 0.01), and C/N (*p* < 0.001) and significantly negatively correlated with TN (*p* < 0.05), AP (*p* < 0.01), and WC (*p* < 0.01).

### 3.6. Soil Bacterial Function Prediction

FAPROTAX was used to predict and annotate the function of soil bacterial community, and there were 12 functional groups with relative abundance greater than 1% (Table 5). Chemoheterotrophy and aerobic chemoheterotrophy were the major bacterial functional groups. Except for predatory or exoparasitic and phototrophy, other bacterial functional groups were significantly different (*p* < 0.05). Among them, chemoheterotrophy and aerobic chemoheterotrophy decreased significantly with the increase in *Larix gmelinii* important values, while human pathogens all and human pathogens pneumonia were increased significantly.

## 4. Discussion

### 4.1. Differences in Soil Physicochemical Properties of Taiga Forests with Different Important Values of Larix gmelinii

Habitat heterogeneity denotes the uneven distribution of numerous biotic and abiotic elements that influence ecological processes within the natural environment [27]. Soil performs a vital role in forest ecosystems and can display strong heterogeneity due to changes in the aboveground vegetation communities [28]. This study noted that taiga forests with different important values of *Larix gmelinii* exhibited heterogeneous habitats, and soil pH, SOC, MBC, and C/N ratio decreased significantly as the important values of *Larix gmelinii* increased. The noticeable decrease in soil pH can be attributed to various factors, including the decomposition of coniferous litter, the production of root exudates, and bacterial metabolism. These processes have been found to generate organic acids, which in turn contribute to the reduction in soil pH [29,30]. The decline in soil SOC and MBC may be related to the higher amounts of recalcitrant substances in needle litter of taiga forests, like cellulose and lignin. These substances inhibit the return of nutrients to the soil and reduce the availability of microbial substrates, thereby limiting organic carbon accumulation and reducing microbial abundance and activity [31,32,33,34]. A decrease in soil C/N ratio may result from *Larix gmelinii* roots secreting substantial quantities of organic matter and releasing more nitrogen by degradation, which affects the decomposition of organic matter and the mineralization ability of soil nitrogen [35]. Moreover, this study also found that the content of AP increased significantly as important values rose. The reason may be that soil phosphorus solubilizing bacteria (e.g., Proteobacteria, Actinobacteria, Bacteroidetes, etc.) can directly or indirectly activate insoluble phosphorus, increasing the effective phosphorus content of soil [36,37].

### 4.2. The Heterogeneous Habitat in Taiga Forests Changed the Diversity and Community Composition of Soil Bacteria

Habitat is the specific environment upon which organisms rely for survival, and its heterogeneity is crucial to maintaining biodiversity [38]. According to this study, the alpha diversity of soil bacteria did not clearly follow changes in the important values of *Larix gmelinii*, but some experimental groups showed significant differences. The potential reasons for these differences may be that the soil carbon, nitrogen, and other nutrient contents in the heterogeneous habitats of taiga forests were significantly different, which directly impacted the quantity and spatial distribution of soil bacterial communities [39,40]. Additionally, the distinct vegetation community structures of taiga forests may cause differences in secondary metabolite (e.g., organic acids, sugars, phenols, etc.) input from plant roots to soil. These differences may also contribute to varying degrees of changes in soil bacterial diversity [41]. However, further research and investigation are needed to fully understand these factors.

This study also found that soil bacterial beta diversity significantly changed in taiga forests due to their heterogeneous habitats. RDA analysis showed that the soil bacterial community structure was driven and regulated by several soil physicochemical factors, including SOC, C/N, MBC, AP, TN, pH, AN, and WC, which was consistent with most previous studies [42,43,44]. The soil nutrient contents exhibited notable variations within the heterogeneous habitat of taiga forests. Soil carbon, nitrogen, and phosphorus are necessary nutrients that serve as an essential part in supporting microbial growth and metabolic processes. Consequently, alterations in the contents of these nutrients may immediately influence the beta diversity of soil bacteria [45]. Meanwhile, soil pH is believed to be a key factor determining soil bacterial communities, and distinct bacterial species are capable of tolerating different pH levels [46]. Thus, the significant decrease in soil pH greatly impacted bacterial community structure in this study. Furthermore, soil moisture content was another critical driver of the bacterial beta diversity. Soil bacteria are more sensitive to soil moisture changes, and both excessively high and low soil moisture content are detrimental to their growth [44]. Therefore, the shifts in soil moisture content showed a substantial influence on soil bacteria in this study.

Habitat heterogeneity is a fundamental characteristic of forest ecosystems, and it has many effects on microorganisms. It not only changes microbial diversity but also influences the composition of bacterial communities [47,48]. Proteobacteria, Actinobacteriota, and Acidobacteriota were the dominant bacterial phyla in the taiga forests, which was consistent with the result of a previously related report [49]. This is mostly due to Proteobacteria, Actinobacteriota, and Acidobacteriota having a wide ecological range and being more adaptable to external environmental changes [50], so they are less affected by the heterogeneous habitat of taiga forests. Bacteroidota, Myxococcota, and Chloroflexi were the main bacterial phyla in the taiga forests, and their relative abundances showed distinctive trends as important values of *Larix gmelinii* increased. Among them, there was a discernible decrease in the relative abundances of Bacteroidota and Myxococcota, which presented a significant positive correlation with SOC, MBC, and C/N. It can be explained by Bacteroidota and Myxococcota as eutrophic bacteria that grow rapidly under nutrient-rich conditions [51]. Consequently, the pronounced decline in SOC, MBC, and C/N within the heterogeneous habitat of taiga forests imposed an inhibiting influence on their growth. Furthermore, previous research has proposed that Bacteroidota exhibits optimal growth conditions in alkaline soils [52]. However, the pH of the taiga forest ranged from 4.31 to 5.48 and showed a considerable reduction in this study, which might also have a negative effect on Bacteroidota. In contrast, the relative abundance of Chloroflexi showed a rising trend. This could potentially be linked to their unique photosynthetic traits, primarily utilizing CO2 as a carbon source to generate energy, so that it can occupy a certain advantage in soils with significantly reduced organic carbon content [53]. At the genus level, *Mycobacterium* was the dominant bacterial genus in the taiga forests, mainly due to its characteristics of being a heterotrophic nitrifying bacterium with fast growth rate, acid tolerance, and strong environmental adaptability [54]. Meanwhile, as one of the main bacterial genera in the taiga forests, *Bradyrhizobium* showed a significant increase as the important values of *Larix gmelinii* rose and was significantly negatively correlated with pH. *Bradyrhizobium*, a type of heterotrophic microbe, exhibits a preference for acidic environments [55]. The significant decline in soil pH created ideal conditions for the growth and reproduction of *Bradyrhizobium*, thus resulting in an increase in its relative abundance in this study.

### 4.3. The Heterogeneous Habitat of Taiga Forest Changed the Potential Functional Groups of Soil Bacteria

Soil bacteria have diverse ecological functions and are an important part of soil nutrient cycling. The FARPOTAX well-revealed the response of soil bacterial function to the variations in constructive species in the taiga forests. The findings of this study indicate that the predominant ecological functions of soil bacteria were chemoheterotrophy and aerobic chemoheterotrophy. Prior study has demonstrated that chemoheterotrophy and aerobic chemoheterotrophy are common ecological functions that are intricately linked to the carbon cycling [56], which is consistent with the findings of this study. The results from the one-way analysis of variance revealed a significant reduction in the relative abundance of chemoorganotrophic and aerobic chemoheterotrophic bacteria as the important value of *Larix gmelinii* grew. This decrease might be assigned to the inhibition of growth and distribution of chemoorganotrophic and aerobic chemoheterotrophic bacteria, which may be caused by a substantial drop in soil SOC [57]. In addition, the slow decomposition of coniferous litter in taiga forests leads to its extensive accumulation on the soil surface, resulting in the formation of a microenvironment with poor air circulation [50], which may also have an adverse effect on aerobic heterotrophic bacteria.

## 5. Conclusions

(1)There were heterogeneous habitats in taiga forests with different important values of *Larix gmelinii*, in which AP content increased significantly as the important value rose, while SOC, MBC, pH, and C/N decreased significantly.(2)The composition of soil bacterial community in the heterogeneous habitat of taiga forests was similar. Proteobacteria, Actinobacteriota, and Acidobacteriota were the dominant bacterial phyla with no significant difference; *Mycobacterium* was the dominant bacterial genus, and the relative abundance of each bacterial group was different.(3)The beta diversity of soil bacterial communities in taiga forests showed highly significant differences, with SOC, C/N, MBC, AP, TN, pH, AN, and WC being important influencing factors for their significant changes.(4)Chemoheterotrophy and aerobic chemoheterotrophy were the main functional groups of soil bacteria, and the relative abundance of each bacterial functional group was significantly different.

## Figures and Tables

**Figure 1 microorganisms-11-02609-f001:**
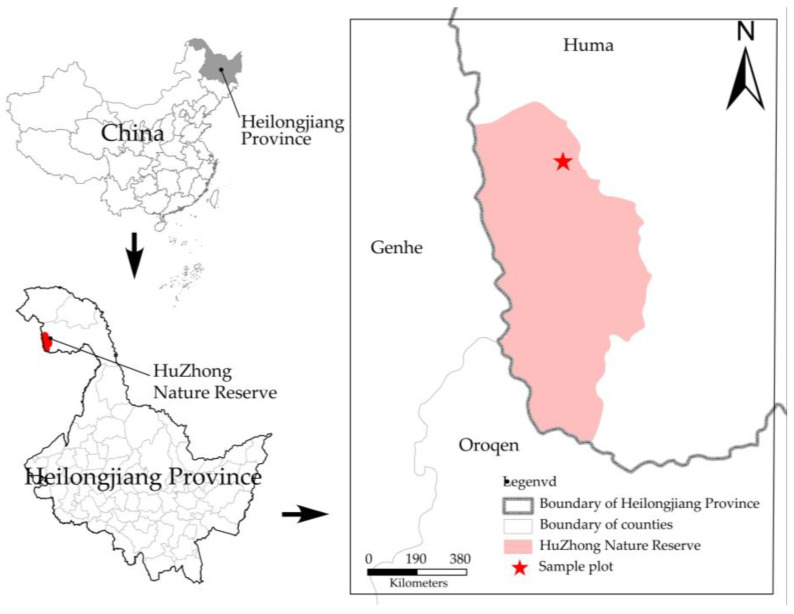
Overview of the Greater Khingan Research Area. The asterisk indicates the study site in the Huzhong National Nature Reserve, Heilongjiang Province, China.

**Figure 2 microorganisms-11-02609-f002:**
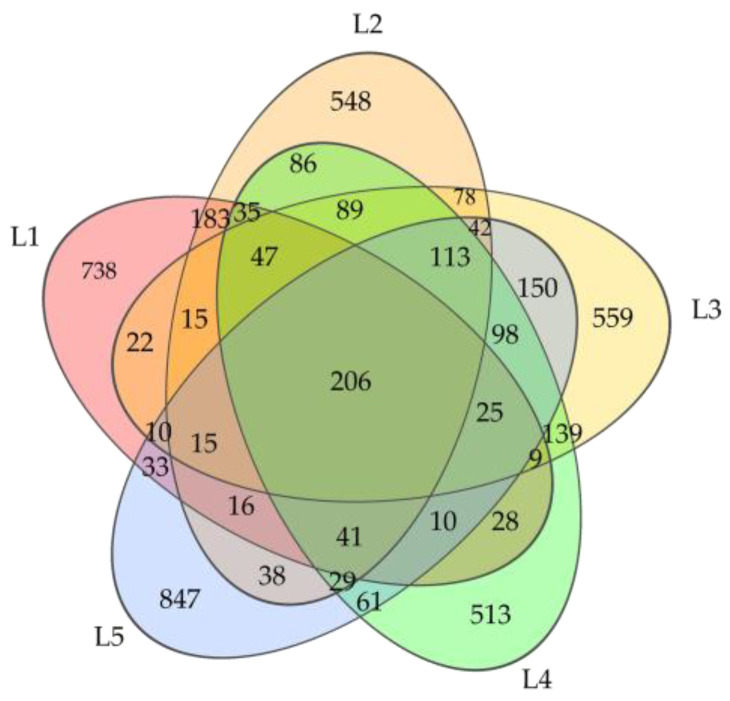
The Venn diagram of soil bacteria ASVs in taiga forests with different important values.

**Figure 3 microorganisms-11-02609-f003:**
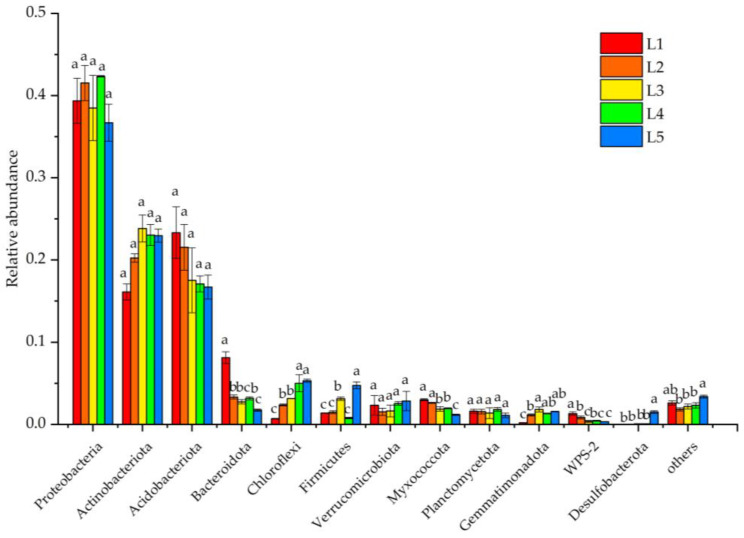
Composition of soil bacterial communities at the phylum level in taiga forests with different important values.

**Figure 4 microorganisms-11-02609-f004:**
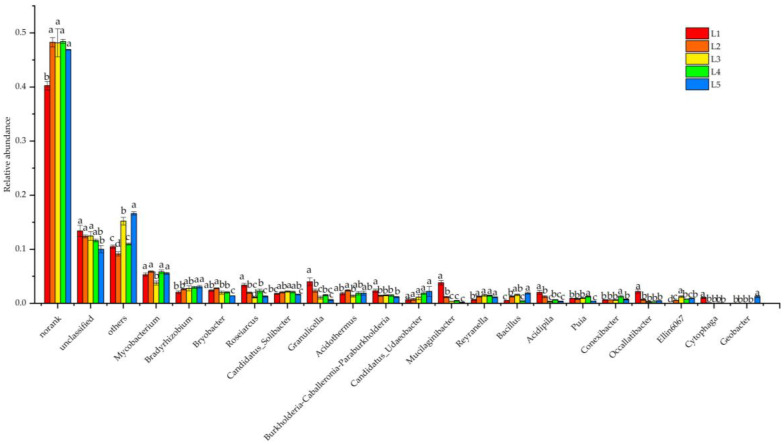
Composition of soil bacterial communities at the genus level in taiga forests with different important values.

**Figure 5 microorganisms-11-02609-f005:**
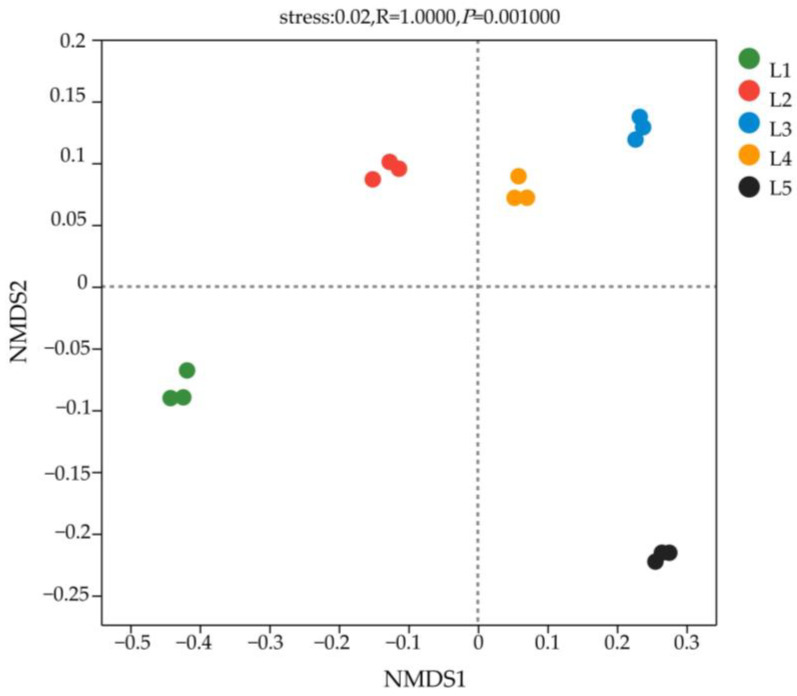
NMDS analysis of soil bacterial communities based on ASV level in taiga forests with different important values.

**Figure 6 microorganisms-11-02609-f006:**
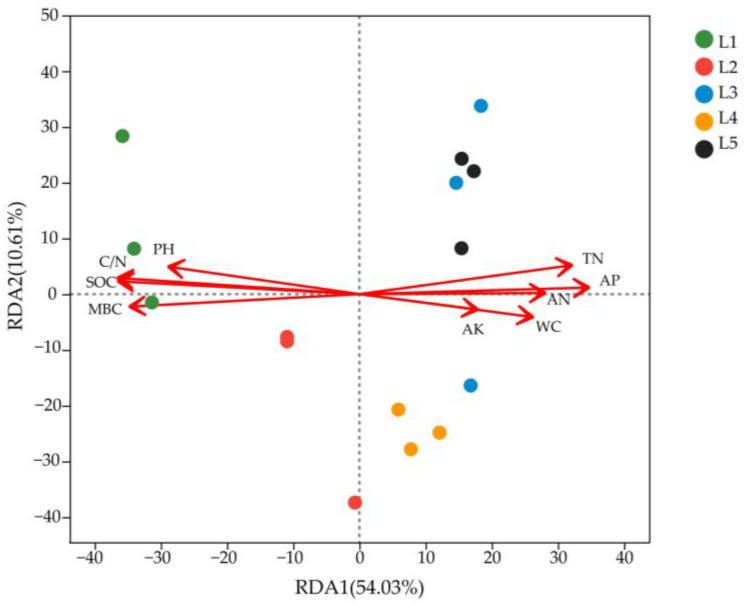
RDA analysis of soil bacterial communities at the genus level in taiga forests with different important values.

**Figure 7 microorganisms-11-02609-f007:**
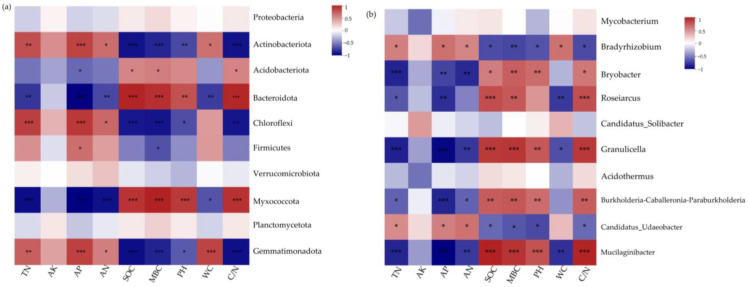
Correlation heatmap of the relative abundance of the top ten soil bacteria and soil physicochemical properties. Color represents correlation strength, with red for positive and blue for negative correlations. * 0.01 < *p* ≤ 0.05, ** 0.001 < *p* ≤ 0.01, *** *p* ≤ 0.001. (**a**) is the correlation between soil bacteria and soil physicochemical factors at the phylum level, (**b**) is the correlation between soil bacteria and soil physicochemical factors at the genus level).

**Table 1 microorganisms-11-02609-t001:** The basic situation of the important value of *Larix gmelinii* in each experimental group. The data in the table are mean ± standard error (*n* = 3), and different lowercase letters in the same column indicate significant difference (*p* < 0.05).

Experimental Group	Important Value
L1	0.434 ± 0.009 ^e^
L2	0.481 ± 0.007 ^d^
L3	0.514 ± 0.012 ^c^
L4	0.544 ± 0.007 ^b^
L5	0.585 ± 0.009 ^a^

**Table 2 microorganisms-11-02609-t002:** Differences in soil physicochemical properties of taiga forests with different important values. The data in the table are mean ± standard error (*n* = 3), and different lowercase letters in the same column indicate significant difference (*p* < 0.05).

Experimental Group	TN (g/kg)	AK (mg/kg)	AP (mg/kg)	AN (mg/kg)	SOC (g/kg)	MBC (mg/kg)	pH	WC (%)	C/N
L1	2.14 ± 0.02 ^b^	137.44 ± 0.92 ^c^	20.23 ± 0.08 ^e^	67.57 ± 0.13 ^c^	73.66 ± 1.46 ^a^	200.77 ± 0.09 ^a^	5.48 ± 0.01 ^a^	31.16 ± 1.07 ^b^	34.39 ± 0.73 ^a^
L2	2.15 ± 0.02 ^b^	112.21 ± 0.68 ^e^	45.25 ± 2.02 ^d^	69.47 ± 0.17 ^c^	55.54 ± 0.33 ^b^	170.70 ± 0.06 ^b^	5.14 ± 0.01 ^b^	34.77 ± 0.53 ^ab^	25.80 ± 0.29 ^b^
L3	2.27 ± 0.01 ^a^	191.49 ± 0.97 ^a^	57.16 ± 0.65 ^c^	79.68 ± 1.46 ^b^	42.18 ± 0.53 ^c^	147.77 ± 0.03 ^c^	5.05 ± 0.00 ^c^	36.35 ± 2.35 ^a^	18.58 ± 0.27 ^c^
L4	2.29 ± 0.00 ^a^	188.15 ± 0.62 ^b^	61.56 ± 1.34 ^b^	96.13 ± 1.17 ^a^	40.29 ± 0.11 ^cd^	139.67 ± 0.15 ^d^	4.41 ± 0.01 ^d^	36.06 ± 1.18 ^a^	17.59 ± 0.05 ^cd^
L5	2.31 ± 0.01 ^a^	131.45 ± 0.70 ^d^	77.38 ± 0.75 ^a^	98.35 ± 0.70 ^a^	39.41 ± 0.22 ^d^	116.40 ± 0.20 ^e^	4.31 ± 0.00 ^e^	36.86 ± 1.08 ^a^	17.09 ± 0.07 ^d^

**Table 3 microorganisms-11-02609-t003:** Alpha diversity of soil bacteria communities in taiga forests with different important values. The data in the table are mean ± standard error (*n* = 3), and different lowercase letters in the same column indicate significant difference (*p* < 0.05).

Experimental Group	Coverage Index	Chao1 Index	Shannon Index	Simpson Index
L1	0.9996 ± 0.0002 ^a^	861.4867 ± 45.3938 ^a^	5.9228 ± 0.0613 ^b^	0.0085 ± 0.0011 ^a^
L2	0.9997 ± 0.0001 ^a^	896.1055 ± 29.5726 ^a^	6.0373 ± 0.0434 ^ab^	0.0077 ± 0.0005 ^ab^
L3	0.9998 ± 0.0001 ^a^	902.6085 ± 46.4267 ^a^	6.1131 ± 0.0958 ^ab^	0.0068 ± 0.0012 ^ab^
L4	0.9999 ± 0.0001 ^a^	866.5270 ± 34.9195 ^a^	5.9670 ± 0.0207 ^b^	0.0078 ± 0.0005 ^ab^
L5	0.9998 ± 0.0001 ^a^	966.1622 ± 51.2490 ^a^	6.2130 ± 0.0616 ^a^	0.0055 ± 0.0006 ^b^

**Table 4 microorganisms-11-02609-t004:** Significance tests between soil bacterial community and soil physicochemical properties.

Soil Factors	R^2^	*p*-Value
TN	0.7442	0.003
AK	0.2187	0.224
AP	0.8497	0.001
AN	0.5492	0.009
SOC	0.9429	0.001
MBC	0.8539	0.001
pH	0.5905	0.004
WC	0.4834	0.019
C/N	0.9381	0.001

**Table 5 microorganisms-11-02609-t005:** Differences in potential functional groups with average relative abundance > 1% in different important values of taiga forests. The data in the table are mean ± standard error (*n* = 3), and different lowercase letters in the same column indicate significant difference (*p* < 0.05).

Functional Groups	L1	L2	L3	L4	L5
chemoheterotrophy	40.6 ± 0.11 ^a^	38.93 ± 0.20 ^b^	37.02 ± 0.54 ^c^	38.68 ± 0.35 ^b^	35.08 ± 0.09 ^d^
aerobic chemoheterotrophy	40.45 ± 0.12 ^a^	38.87 ± 0.19 ^b^	36.87 ± 0.48 ^c^	38.54 ± 0.36 ^b^	33.59 ± 0.21 ^d^
nitrogen fixation	4.44 ± 0.50 ^b^	5.78 ± 0.31 ^a^	6.24 ± 0.51 ^a^	6.59 ± 0.32 ^a^	6.14 ± 0.43 ^a^
cellulolysis	6.14 ± 0.25 ^a^	5.58 ± 0.43 ^ab^	3.31 ± 0.43 ^c^	4.52 ± 0.86 ^abc^	3.72 ± 0.72 ^bc^
animal parasites or symbionts	1.50 ± 0.55 ^b^	1.50 ± 0.19 ^b^	2.23 ± 0.19 ^ab^	1.91 ± 0.24 ^ab^	2.78 ± 0.45 ^a^
predatory or exoparasitic	1.44 ± 0.06 ^a^	1.53 ± 0.07 ^a^	1.53 ± 0.15 ^a^	1.73 ± 0.09 ^a^	1.92 ± 0.36 ^a^
human pathogens all	0.36 ± 0.01 ^d^	0.65 ± 0.09 ^cd^	1.54 ± 0.09 ^b^	0.95 ± 0.16 ^c^	1.94 ± 0.08 ^a^
human pathogens pneumonia	0.09 ± 0.03 ^e^	0.62 ± 0.08 ^d^	1.54 ± 0.09 ^b^	0.95 ± 0.16 ^c^	1.91 ± 0.09 ^a^
ureolysis	0.39 ± 0.01 ^d^	0.43 ± 0.06 ^cd^	1.93 ± 0.08 ^a^	0.78 ± 0.12 ^c^	1.22 ± 0.22 ^b^
phototrophy	1.01 ± 0.06 ^a^	0.95 ± 0.05 ^a^	0.93 ± 0.18 ^a^	0.81 ± 0.11 ^a^	0.80 ± 0.05 ^a^
iron respiration	0.00 ± 0.00 ^b^	0.00 ± 0.00 ^b^	0.05 ± 0.03 ^b^	0.00 ± 0.00 ^b^	2.42 ± 0.32 ^a^
fermentation	0.04 ± 0.01 ^b^	0.03 ± 0.02 ^b^	0.12 ± 0.06 ^b^	0.07 ± 0.01 ^b^	1.52 ± 0.21 ^a^

## Data Availability

Data available upon request.

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
