# Peer review of "Heterogeneous Habitats in Taiga Forests with Different Important Values of Constructive Species Changes Bacterial Beta Diversity"

_microorganisms, 2023, doi:10.3390/microorganisms11102609_

Round 1
Reviewer 1 Report
The article makes a great contribution to the understanding of the mechanisms of beta diversity of bacterial communities in forest biocenoses. The conclusions are well reasoned, the results were obtained using modern methods and well statistically processed. Contemporary literature relevant to the research topic is cited. The article can be recommended for publication.
Author Response
Dear reviewer:
Thank you very much for taking the time to review this manuscript. We have studied comments carefully, and all of these comments are valuable for our paper.
Thank you again for your kind and thoughtful comments.
Sincerely yours,
Tian Zhou
Reviewer 2 Report
This research paper is scientifically sound and merit to be published in its present form.The English scientific body is well prepared and the results well discussed in depth.I do not find grammatical errors through the study.
Experimental design is appropriate and relevant.Bibliography is up to date Concluding it is an interesting paper which could bring new knowledge to the field.
1. What is the main question addressed by the research?
The authors try to explain the interconnection in the different components of the taiga forest ecosystems between soil bacterial communities and plant communities following soil conditions.
2. Do you consider the topic original or relevant in the field? Does it
address a specific gap in the field?
It is an interesting topic and such a knowledge will help to protect effectively the forests
3. What does it add to the subject area compared with other published
material?
There is little information in these type of interconnections in the literature in my knowledge.
4. What specific improvements should the authors consider regarding the
methodology? What further controls should be considered?
The methodology is well designed .Material and Methods are explained in depth (areas,sampling) .They proceed to a physicochemical analysis of the components ,followed by sequencing .
5. Are the conclusions consistent with the evidence and arguments presented
and do they address the main question posed?
Results were evaluated by bioinformatics and advanced statistical analyses and provide consistent conclusions that were extensively discussed
6. Are the references appropriate?
Their conclusions were extensively discussed based on an extended up to date bibliography
7. Please include any additional comments on the tables and figures.
Tables and figures are of good quality
Author Response
Dear reviewer:
I wanted to express my gratitude for your time and effort in reviewing our manuscript. Your feedback and suggestions have been incredibly valuable in improving the quality of our work.
I understand from your review that you did not have any specific concerns or comments regarding the manuscript. I want to assure you that we have carefully reviewed the paper ourselves and have made necessary revisions to ensure its accuracy and clarity.
Once again, thank you for your diligent review. Your input has significantly contributed to the refinement of our manuscript.
Best regards,
Tian Zhou
Reviewer 3 Report
The manuscript presents the studies that were analyzed the vegetation within 25 hm2 monitoring plot in cold-temperate coniferous forests, and estimated the important value of Larix gmelinii to reflect the differences of constructive species. The soil of taiga forests with different important values were taken as the study objects, and the high-throughput sequencing technology was employed to examine the difference of community composition and diversity of soil bacteria, as well as the factors influencing these variations.
The manuscript is consistent with the profile of the journal. The research presented in it was carried out methodologically correctly. The results were well presented, statistically analyzed and clearly described. The Discussion chapter has been presented correctly. The conclusions are well formulated and result from the scope of the research performed.
In my opinion, the manuscript is suitable for publication after minor correction of the reference list, because not all literature items are cited according to the same pattern. This includes items 4, 6, 47, 52 and 53.
It is a pity that the authors did not take soil grain size into account in their research, because soil grain size is closely related to the physicochemical properties analyzed by the authors of the manuscript.
